

# Quantile-dependent expressivity of serum C-reactive protein concentrations in family sets

Paul T. Williams

Molecular Biophysics & Integrated Bioimaging Division, Lawrence Berkeley National Laboratory, Berkeley, CA, USA

## ABSTRACT

**Background:** "Quantile-dependent expressivity" occurs when the effect size of a genetic variant depends upon whether the phenotype (e.g., C-reactive protein, CRP) is high or low relative to its distribution. We have previously shown that the heritabilities ($h^2$) of coffee and alcohol consumption, postprandial lipemia, lipoproteins, leptin, adiponectin, adiposity, and pulmonary function are quantile-specific. Whether CRP heritability is quantile-specific is currently unknown.

**Methods:** Serum CRP concentrations from 2,036 sibships and 6,144 offspring-parent pairs were analyzed from the Framingham Heart Study. Quantile-specific heritability from full-sib ($\beta_{FS}$, $h^2 = \{(1 + 8r_{spouse}\beta_{FS})^{0.5} - 1\}/(2r_{spouse})$) and offspring-parent regression slopes ($\beta_{OP}$, $h^2 = 2\beta_{OP}/(1 + r_{spouse})$) were estimated robustly by quantile regression with nonparametric significance determined from 1,000 bootstrap samples.

**Results:** Quantile-specific $h^2$ (±SE) increased with increasing percentiles of the offspring's age- and sex-adjusted CRP distribution when estimated from $\beta_{OP}$ ($P_{trend} = 0.0004$): 0.02 ± 0.01 at the 10th, 0.04 ± 0.01 at the 25th, 0.10 ± 0.02 at the 50th, 0.20 ± 0.05 at the 75th, and 0.33 ± 0.10 at the 90th percentile, and when estimated from $\beta_{FS}$ ($P_{trend} = 0.0008$): 0.03±0.01 at the 10th, 0.06 ± 0.02 at the 25th, 0.14 ± 0.03 at the 50th, 0.24 ± 0.05 at the 75th, and 0.53 ± 0.21 at the 90th percentile.

**Conclusion:** Heritability of serum CRP concentration is quantile-specific, which may explain or contribute to the inflated CRP differences between *CRP* (rs1130864, rs1205, rs1800947, rs2794521, rs3091244), *FGB* (rs1800787), *IL-6* (rs1800795, rs1800796), *IL6R* (rs8192284), *TNF-α* (rs1800629) and *APOE* genotypes following CABG surgery, stroke, TIA, curative esophagectomy, intensive periodontal therapy, or acute exercise; during acute coronary syndrome or *Staphylococcus aureus* bacteremia; or in patients with chronic rheumatoid arthritis, diabetes, peripheral arterial disease, ankylosing spondylitis, obesity or inflammatory bowel disease or who smoke.

Corresponding author
Paul T. Williams,
1742spyglass@comcast.net

## INTRODUCTION

C-reactive protein (CRP) concentrations are reflective of low-grade systemic inflammation. Higher basal concentrations are associated with increasing age, obesity, smoking, disease (Alzheimer's, cardiovascular, Type 2 diabetes mellitus (T2DM)) and female sex (*Windgassen et al., 2011*). Prospectively, plasma CRP-concentrations predict de novo atherothrombotic cardiovascular events (*Kaptoge et al., 2010*). Basal CRP concentrations are also in part genetic, with an estimated heritability of about 35%, but with individual estimates varying greatly (*Sas et al., 2017*). CRP concentration may increase by 500-fold following an acute-phase stimulus due to enhanced hepatic transcription, primarily in response to the proinflammatory cytokine interleukin 6 (*Pepys, 2003*; *Hage & Szalai, 2007*). Clinically, CRP concentrations are used for the diagnosis and monitoring of inflammatory processes (*Pepys, 2003*) in rheumatologic disease (*Rhodes et al., 2010*; *Ammitzbøll et al., 2014*; *Wielińska et al., 2020*), ankylosing spondylitis (*Xu, Jiang & Zhang, 2020*), inflammatory bowel disease (*Vatay et al., 2003*), pancreatitis (*Windgassen et al., 2011*), cardiovascular disease (*Hage & Szalai, 2007*; *Ni et al., 2020*), cancer (*Windgassen et al., 2011*), and infections (*Pepys, 2003*).

"Quantile-dependent expressivity" postulates that the effects of genetic variants on phenotypes may depend on the whether the phenotype (e.g., CRP concentration) is high or low relative to its distribution. The heritability of adiposity (*Williams, 2012*, *2020a*); plasma concentrations of triglyceride (*Williams, 2012*, *2020b*), high-density lipoproteins (*Williams, 2012*, *2020c*, *2020d*), total cholesterol (*Williams, 2020e*), leptin (*Williams, 2020f*), and adiponection (*Williams, 2020g*); pulmonary function (*Williams, 2020h*); and intakes of alcohol (*Williams, 2020i*) and coffee (*Williams, 2020j*) are quantile dependent, whereas height and the intakes of other macronutrients are not (*Williams, 2012*, *2020a*, *2020i*). Others have also demonstrated increasing genetic effect size with increasing BMI levels (*Rokholm et al., 2011*; *Abadi et al., 2012*; *Beyerlein et al., 2011*; *Mitchell et al., 2013*). A particularly compelling case for quantile-dependent expressivity is the linear increases in the effect sizes of single nucleotide polymorphisms (SNP) with postprandial increases in triglyceride (*Williams, 2020k*) and adiponectin (*Williams, 2020g*) concentrations during lipemia–compelling because their concordant increases are demonstrable within individuals and within hours, exclusive of other sources of temporal and between-subject variation. Many purported examples of gene-environment interactions may be attributable to quantile-dependent expressivity when subjects are selected for conditions that distinguish high vs. low phenotype values (*Williams, 2020d*). With respect to precision-medicine, genetic markers for identifying patients most likely to benefit from medications or diet may also be artifacts of quantile-dependent expressivity when the markers simply track the change in heritability associated with drug-, diet-, or behavior-induced changes in the average phenotype value (*Williams, 2020b*, *2020c*, *2020e*, *2020k*).

It is not known whether CRP heritability is quantile specific or whether the CRP gene-environment interactions reported by others are consistent with quantile-dependent expressivity. Therefore, quantile-dependent expressivity of CRP was investigated by

applying quantile regression (*Koenker & Hallock, 2001*; *Gould, 1992*) to sibships and offspring-parent pairs from the Framingham Heart Study (*Kannel et al., 2006*; *Splansky et al., 2007*) to estimate heritability in the narrow sense ($h^2$ (*Falconer & Mackay, 1996*)) at different quantile of the CRP distributions. Heritability of untransformed CRP concentrations was studied because only a small proportion of CRP variation is attributable to specific SNPs (*Dehghan et al., 2011*), because quantile regression does not require statistical normality (*Koenker & Hallock, 2001*; *Gould, 1992*), and because no biological justification for logarithmic transforming CRP concentrations has heretofore been provided. The discussion furthers this investigation by re-examining published examples of CRP gene-environment interactions from the perspective of quantile-dependent expressivity. Of particular interest are the effects of genetic variants on CRP concentrations during its acute phase response to infections, trauma, and surgery because these may exceed basal CRP levels by over 100-fold (*Pepys, 2003*; *Agrawal, 2005*; *Danik & Ridker, 2007*). Quantile-dependent expressivity hypothesizes that genetic effects on CRP concentrations should increase in accordance with changing CRP concentrations during intermediate and peak increases in it's acute phase concentrations.

## METHODS

The Framingham Study data were obtained from the National Institutes of Health FRAMCOHORT, GEN3, FRAMOFFSPRING Research Materials obtained from the National Heart, lung, and Blood (NHLBI) Biologic Specimen and Data Repository Information Coordinating Center. The hypothesis tested not considered as part of the initial Framingham Study design and is exploratory. The Framingham Heart Study included three cohorts. The Original Cohort includes 5,209 30–59 year old men and women who lived in Framingham, Massachusetts. The Offspring Cohort is made up of the 5,124 adult children of the Original Cohort and their spouses. They were initially examined between 1971 and 1975, reexamined 8 years later, and then every 3–4 years thereafter (*Kannel et al., 2006*). The Third Generation Cohort is the children of the Offspring Cohort (*Splansky et al., 2007*). Subjects used in the current analyses were at least 16 years of age and were self-identified as non-Hispanic white. Phlebotomy was performed on fasting participants who had rested for 5–10 min in a supine position, typically between 8 and 9 AM. Specimens were stored at −80 °C without freeze-thaw cycles until assay. Serum high-sensitivity CRP concentrations were measured with a Dade Behring BN100 nephelometer (Deerfield, IL, USA) with a Kappa statistic of 0.95 for 146 samples run in duplicate (*Shoamanesh et al., 2015*). Plasma CRP concentrations were measured for examinations 2, 6, 7, 8, and 9 of the Offspring Cohort, and examinations 1 and 2 of the Third Generation Cohort.

Our analyses of these data were approved by Lawrence Berkeley National Laboratory Human Subjects Committee (HSC) for protocol "Gene-environment interaction vs. quantile-dependent penetrance of established SNPs (107H021)." LBNL holds Office of Human Research Protections Federal wide Assurance number FWA 00006253. Approval number: 107H021-13MR20. Signed informed consent were obtained from all participants

or parent and/or legal guardian if <18 years of age. All surveys were conducted under the guidelines set forth by the Framingham Heart Study human use committee.

## Statistics

The statistical methodology has been described in detail elsewhere (*Williams, 2012*, *2020a*, *2020b*, *2020c*, *2020d*, *2020e*, *2020f*, *2020g*, *2020h*, *2020i*, *2020j*, *2020k*) and is summarized here briefly for completeness. The only eligibility requirement for inclusion in the analyses was CPR values for offspring, parents and siblings. Standard least-squares regression was used for sex and age adjustment separately in each cohort using female (0,1), age, $age^2$, female $\times$ age, and female $\times$ $age^2$ as independent variables. Individual subject CRP values were taken as the average over all available exams of the age and sex-adjusted concentrations. Parents from the Offspring Cohort and their children from the Third Generation Cohort were used to compute offspring-parent regression slopes ($\beta_{OP}$). Siblings were obtained from the Third Generation and Offspring Cohorts. Full-sibling regression slopes ($\beta_{FS}$) were calculated by forming all $k_i(k_i - 1)$ sibpair combinations for the $k_i$ siblings within sibship $i$ and assigning equal weight to each sibling (*Karlin, Cameron & Williams, 1981*).

The sqreg command of Stata (version. 11; StataCorp, College Station, TX, USA) was used to perform simultaneous quantile regression. The variance-covariance matrix for the ninety-one quantile regression coefficients between the 5th and 95th percentiles of the offspring's distribution was estimated from 1,000 bootstrap samples (*Gould, 1992*). The test and lincom post-estimation procedures were used to test linear combinations of the slopes with $\Sigma(k_i - 1)$ degrees of freedom for sibship regression slopes and $\Sigma k_i - 2$ degrees of freedom for offspring-parent regression slopes. Quantile-specific expressivity was assessed by: (1) estimating the quantile-specific $\beta$-coefficients ($\pm$SE) for the 5th, 6th,…, 95th percentiles of the sample distribution; (2) plotting the quantile-specific $\beta$ coefficient vs. the quantile of the trait distribution; and (3) testing whether the quantile-specific $\beta$-coefficients were constant, or changed as linear, quadratic, or cubic functions of the percentile of the trait distribution using orthogonal polynomials (*Winer, Brown & Michels, 1991*). Falconer and Mackay's formula (*Falconer & Mackay, 1996*) equates narrow-sense heritability ($h^2$) to $h^2 = 2\beta_{OP}/(1 + r_{spouse})$ and to $h^2 = \{(1 + 8\beta_{FS}r_{spouse})^{0.5} - 1\}/2r_{spouse}$ under specific restrictive assumptions, where $r_{spouse}$ is the spouse correlation. "Quantile-dependent expressivity" is the biological phenomenon of the trait expression being quantile-dependent, whereas "quantile-specific heritability" refers to the heritability statistic.

The findings of other studies were analyzed from the perspective of quantile-dependent expressivity from the genotype-specific mean CRP concentrations cited in the original articles or by calculating these values from the published graphs using the formatting palette for Microsoft Powerpoint (Microsoft corporation, Redmond WA, version 12.3.6 for Macintosh computers) as previously employed (*Williams, 2020k*). The weighted average of the geometric means or median values were used to approximate average concentration by condition or pooled genotypes. The interpretations of the current report are not necessarily the same as those of the original articles.
### Data availability

The data are not being published in accordance with the data use agreement between the NIH National Heart Lung, and Blood Institute and Lawrence Berkeley National Laboratory. However, the data used in the analyses are available from NIH National Heart Lung, and Blood Institute Biologic Specimen and Data Repository Information Coordinating Center through the website https://biolincc.nhlbi.nih.gov/my/submitted/request/ (*NIH, 2020*). There are some restrictions to the availability of these data. Researchers wishing a copy of the data should contact the Blood Institute Biologic Specimen and Data Repository Information Coordinating Center at the website provided above, which provides information on human use approval and data use agreement required. The dbGaP study home page (*dbGaP genotypes and phenotypes, 2020*) provides public summary-level phenotype information.

## RESULTS

Eight of the 4,078 offspring in the Third Generation Cohort lacked at least one CRP measurement, and 317 lacked parental information. There was little difference between offspring included vs. excluded from the offspring-parent regression analysis for the proportion of female (mean ± SE: 53.3 ± 0.8 vs. 54.0 ± 2.8%), age (40.1 ± 0.1 vs. 41.2 ± 0.6 years), BMI (27.4 ± 0.1 vs. 28.0 ± 0.3), and CRP concentrations (2.59 ± 0.06 vs. 3.21 ± 0.30 mg/L). Six hundred ninety three participants of the Third Generation Cohort were excluded from the full-sib analysis because they lacked siblings. Again there was little difference between those included vs. excluded from the full-sib regression analysis for the proportion of female (mean ± SE: 53.0 ± 0.9 vs. 55.0 ± 1.9%), age (40.4 ± 0.1 vs. 38.8 ± 0.4 years), BMI (27.4 ± 0.1 vs. 27.4 ± 0.2), and CRP concentrations (2.58 ± 0.07 vs. 2.90 ± 0.17 mg/L).

### Traditional estimates of familial concordance and heritability

Table 1, which displays the sample characteristics, shows that average CRP was significantly higher in women than men. As expected CRP-concentrations were correlated positively with BMI ($r = 0.38$) and were higher in smokers than nonsmokers (difference ± SE: 0.54 ± 0.18, $P = 0.008$) when age and sex adjusted. The spouse correlation for adjusted CRP concentrations was negligible ($r_{spouse} = -0.0013$) for untransformed CRP and weak ($r_{spouse} = 0.0482$) for log CRP. There were 1,718 offspring with one parent and 1,232 offspring with two parents. The offspring-parent regression slope for adjusted CRP concentrations ($\beta_{OP}$ ± SE: 0.06 ± 0.01) corresponds to a heritability ($h^2$) of 0.11 ± 0.02. There were 5,703 full-sibs in 2,036 sibships with age and sex-adjusted CRP concentrations, whose full-sib regression slope ($\beta_{FS}$) was 0.08±0.02, which from Falconer's formula, corresponds to a heritability of $h^2 = 0.15 ± 0.03$. Heritability in female offspring was somewhat greater than in male offspring whether computed from $\beta_{OP}$ (0.13 ± 0.03 vs. 0.08 ± 0.03) or $\beta_{FS}$ (0.20 ± 0.06 vs. 0.10 ± 0.06), but not significantly so. Heritabilities for log CRP derived from $\beta_{OP}$ (0.43 ± 0.03) or $\beta_{FS}$ (0.37 ± 0.03) were consistent with published reports (*Sas et al., 2017*).
**Table 1 Sample characteristics*.**

| | Males | | Females | |
|---|---|---|---|---|
| | Offspring Cohort | Third generation cohort | Offspring Cohort | Third generation cohort |
| Sample size | 1,232 | 1,851 | 1,340 | 2,108 |
| Age, years | 56.54 (8.74) | 40.42 (8.72) | 55.64 (9.10) | 39.99 (8.77) |
| BMI, kg/m$^2$ | 28.08 (3.90) | 28.43 (4.79) | 26.65 (5.22) | 26.50 (6.11) |
| Waist/ht | 0.58 (0.06) | 0.56 (0.07) | 0.56 (0.09) | 0.55 (0.10) |
| CRP mg/L-all | 3.58 (5.83) | 2.15 (3.12) | 3.84 (4.97) | 3.05 (4.57) |
| Waist/ht 1st tertile | 3.25 (7.91) | 1.34 (2.60) | 2.63 (3.72) | 1.37 (2.32) |
| Waist/ht 2nd tertile | 3.16 (4.54) | 1.79 (2.53) | 3.34 (3.37) | 2.58 (3.97) |
| Waist/ht 3rd tertile | 4.22 (5.36) | 3.26 (3.79) | 5.64 (6.55) | 5.23 (5.81) |

Note:
* Mean (SD). BMI, body mass index. CRP, C-reactive protein.

## Quantile-dependent expressivity

Figure 1A presents the offspring-parent regression slopes at the 10th, 25th, 50th, 75th, and 90th percentiles of the offspring's CRP distribution along with their corresponding heritability estimates. The slopes get progressively greater with increasing percentiles of the CRP distribution. At the 90th percentile, heritability was 0.33 or nearly 18-fold greater than the heritability at the 10th percentile ($P_{\text{difference}} = 0.001$). Figure 1B, which presents all slopes between the 5th and 95th percentiles, shows a linear increase in heritability (i.e., slope ± SE: 0.0038 ± 0.0010, $P_{\text{linear}} = 0.0004$) as the percentiles of the offspring's distribution increase. There was no significant evidence of nonlinearity (i.e., $P_{\text{quadratic}} = 0.09$; $P_{\text{cubic}} = 0.31$). Quantile-specific heritabilities were individually significant ($P \leq 0.04$) for all percentiles between the 17th and 92nd percentiles of the offspring's distribution. If the heritabilities over all quantiles were constant, then the line segments would be parallel in Fig. 1A, and the graph in Fig. 1B would show a flat line with zero slope. Figure 1C displays the full-sib quantile regression slopes ($\beta_{\text{FS}}$) and the corresponding estimated $h^2$. Each percent increment in the CRP distribution was associated with a 0.0027 ± 0.0008 increase in the full-sib regression slope ($P_{\text{linear}} = 0.0008$) and a 0.0054 ± 0.0016 increase in heritability.

Figure 2 presents quantile-specific heritability for logarithmically transformed CRP. The transformation replaced the significant linear trend for a quadratic trend showing the greatest heritability near the median and declining heritability moving away from the median when estimated from offspring-parent pairs ($P_{\text{quadratic}} = 0.001$) and full siblings ($P_{\text{quadratic}} = 0.06$).

## DISCUSSION

Our analyses of the Framingham Heart Study provide consistent evidence for quantile-specific heritability of untransformed serum CRP concentrations from both offspring-parent and full-sib age- and sex-adjusted values. Heritability at the 90th percentile of the CRP distribution (0.33 ± 0.10) was 18-fold greater than at the 10th

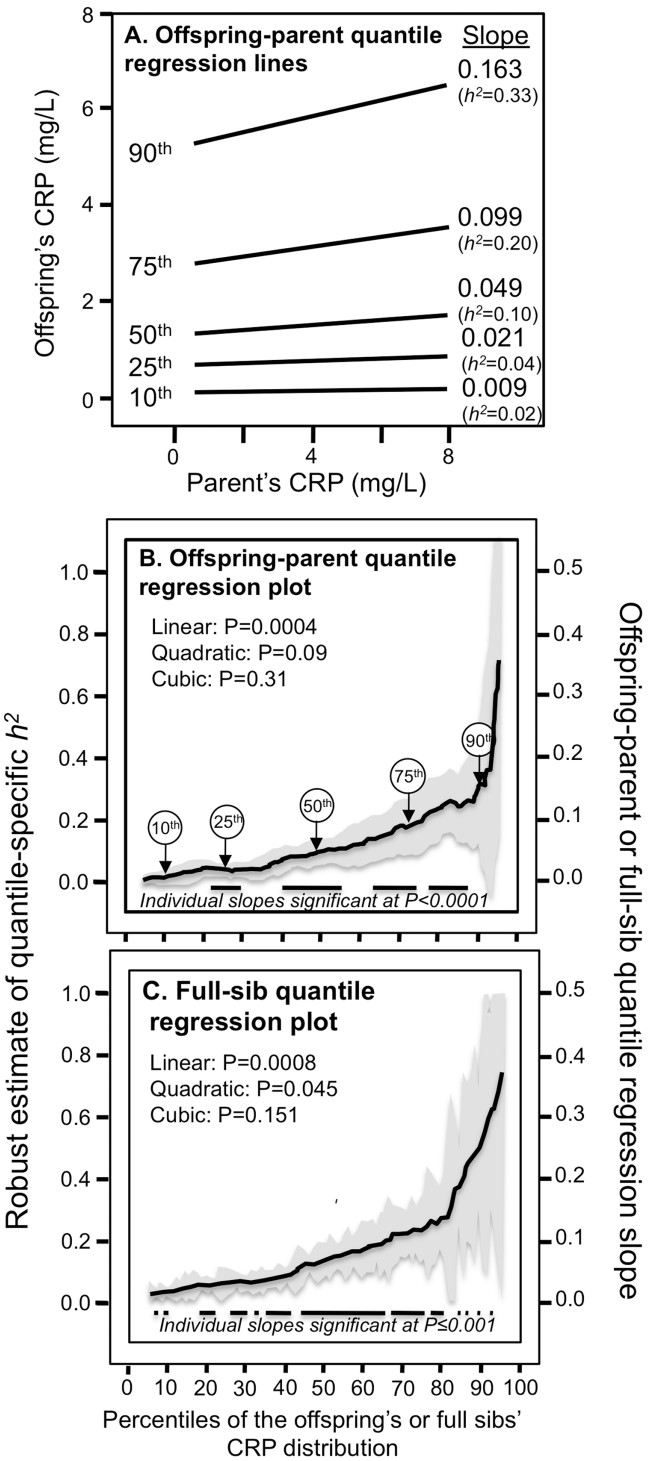

**Figure 1 Quantile-specific offspring-parent ($\beta_{OP}$) and full-sib regression slopes ($\beta_{FS}$) for untransformed CRP concentrations.** (A) Offspring-parent regression slopes ($\beta_{OP}$) for selected quantiles of the offspring's untransformed CRP concentrations from 6,144 offspring-parent pairs, with corresponding estimates of heritability ($h^2 = 2\beta_{OP}/(1 + r_{spouse})$) (*Falconer & Mackay, 1996*), where the correlation between spouses was $r_{spouse} = -0.0013$. The slopes became progressively greater (i.e., steeper) with increasing quantiles of the CRP distribution. (B) The selected quantile-specific regression slopes were included with those of other quantiles to create the quantile-specific heritability function in

**Figure 1 (continued)**
the lower panel. Significance of the linear, quadratic and cubic trends and the 95% confidence intervals
(shaded region) determined by 1000 bootstrap samples. (C) Quantile-specific full-sib regression slopes
($\beta_{FS}$) from 5,703 full-sibs in 2,036 sibships, with corresponding estimates of heritability as calculated by
$h^2 = \{(8r_{spouse}\beta_{FS} + 1)^{0.5} - 1\}/(2r_{spouse})$ (*Falconer & Mackay, 1996*).

percentile ($0.02 \pm 0.01$) when estimated from offspring-parent pairs, and 15-fold greater
when estimated from full sibs. These are substantial differences that exceed those reported
for high-density lipoprotein cholesterol (48% $h^2$ increase in going from the 10th to
90th percentile) (*Williams, 2020c*), adiponectin (72%) (*Williams, 2020g*), total cholesterol
(74%) (*Williams, 2020e*), leptin (4.7-fold greater (*Williams, 2020f*)), or triglycerides
concentrations (13-fold) (*Williams, 2020b*), or BMI (3.1-fold) (*Williams, 2020a*).
We analyzed heritability because it represents 30% to 50% of the CRP additive genetic
variance vis-à-vis the 5% of the CRP variance attributable to 18 specific loci identified by
*Dehghan et al. (2011)* as genomewide significant.

There are, however, important limitations to our analysis of familial phenotypes:
(1) Falconer's formula probably do not adequately address the true complexity of CRP
genetics; and (2) heritability lacks the specificity of directly measured genotypes.
Re-evaluating other published studies that measured genetic variants directly from the
perspective of quantile-dependent expressivity may partly address these concerns.
Consistent with quantile-dependent expressivity, the examples presented below show
larger genetic effect sizes in association with the higher CRP concentrations of low-level
inflammation. Additional examples are presented that suggest the phenomenon may
apply to CRP acute phase reaction. Although several authors do point out that genetic
variant affecting acute phase CRP response are also evident for basal CRP concentrations
(*Rhodes et al., 2010*; *Danik & Ridker, 2007*; *D'Aiuto et al., 2005*), to the best of our
knowledge, quantile-specific heritability has never been formally acknowledged as a
fundamental property of CRP genetics.

The CRP gene is located on chromosome 1q32 and includes two exons and one intron.
*CRP* genetic variants that are reported to affect CRP concentrations include rs2794521
(−717A>G), rs3091244 (−286C>T>A), rs1800947 (+1059G>C), rs1130864 (+1444C>T)
and rs1205 (+2147 A>G). Rs2794521 and rs3091244 are located in the promoter region,
rs1800947 in exon 2, rs1130864 in the 3′ untranslated region, and rs1205 occurs in the
3′ flanking region (*Suk Danik et al., 2006*). Rs3091244 has been shown to affect CRP
transcriptional activity in vitro (*Szalai et al., 2005*). Rs1800947 is silent (*Cao & Hegele,
2000*). Higher basal CRP concentrations are reported for the rs3091244 A-allele, rs1800947
GG-homozygotes, rs1130864 T-allele, and the rs1205 G-allele (*Danik & Ridker, 2007*).
Interleukin-6 (IL-6), the primary inflammatory cytokine stimulus for CRP (*Gabay &
Kushner, 1999*), has two polymorphisms whose minor alleles are reported to increase CRP
concentrations: rs1800795 (−174G>C) (*Vickers et al., 2002*) and rs1800796 (−572G>C)
(*Ferrari et al., 2003*). The tumor necrosis factor α (TNF-α) rs1800629 (G-308A)

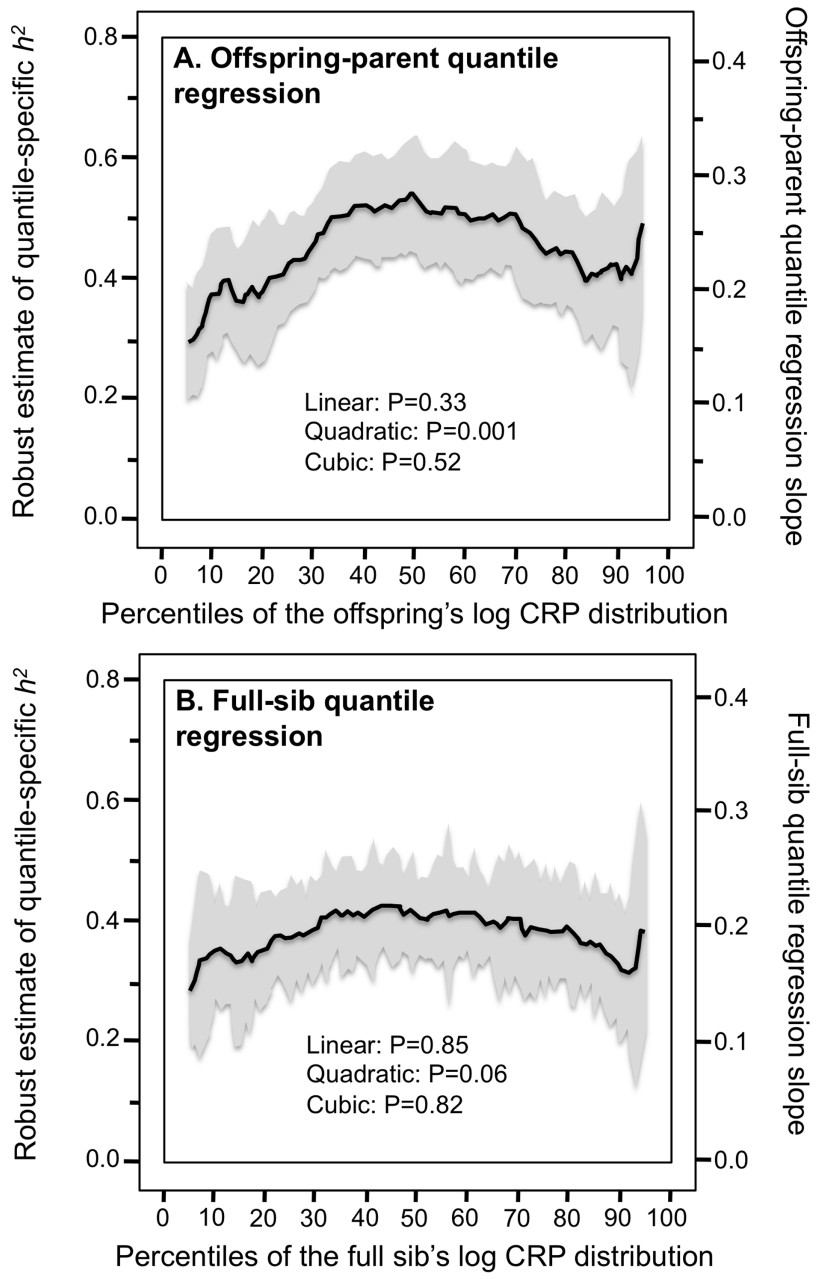

**Figure 2 Quantile-specific offspring-parent ($\beta_{OP}$) and full-sib regression slopes ($\beta_{FS}$) for the offspring's logarithmically transformed CRP concentrations.** (A) Quantile-specific offspring-parent regression slope ($\beta_{OP}$) for the offspring's logarithmically transformed CRP concentrations with corresponding estimates of heritability (*Falconer & Mackay, 1996*), where the correlation between spouses was $r_{spouse} = 0.0482$. (B) full-sib regression slopes ($\beta_{FS}$) for logarithmically transformed CRP concentrations.

polymorphism has been shown to increased TNF-α production in vitro (*Wilson et al., 1997*), and in turn, stimulate hepatic CRP production (*Plutzky, 2001*). Carriers of the APOE ε4 allele have lower CRP-concentrations than non-carriers (*Judson et al., 2004*; *Hubacek et al., 2010*; *Naudé et al., 2018*; *Chasman et al., 2006*).
## Adiposity

BMI, waist circumference and fat body mass are associated with significantly higher CRP concentrations, accounting for five to seven percent of log CRP variation (*Eiriksdottir et al., 2009*). Visceral adipose tissue in particular promotes higher IL-6 concentrations (*Fried, Bunkin & Greenberg, 1998*) and low-grade CRP inflammation (*Forouhi, Sattar & McKeigue, 2001*; *Visser et al., 1999*). CRP concentrations decrease an average of 0.13 mg/L per kg of weight loss (*Selvin, Paynter & Erlinger, 2007*).

Consistent with quantile-dependent expressivity and the higher CRP concentrations of obese subjects, *Friedlander et al. (2006)* reported that the heritability of untransformed CRP was nearly three-fold greater in obese than nonobese subjects (0.670 vs. 0.256). In addition, data reported by *Farup, Rootwelt & Hestad (2020)* showed that the CRP difference between non-carriers and carriers of the *APOE* ε4-allele decreased linearly as average CRP concentrations decreased in morbidly obese patients undergoing weight loss (Fig. 3A). Specifically, the genotype difference was greatest at baseline (ε4- vs. ε4+: 8.2 vs. 5.3 mg/L, $P = 0.004$) when average CRP was highest, intermediate 6 months later (5.2 vs. 3.1 mg/L, $P = 0.007$) for the lower average CRP from losing 3.0 kg/m$^2$ on a conservative weight loss program, and smallest (1.3 vs. 0.7 mg/L, $P = 0.03$) when average CRP was least after losing an additional 10.7 kg/m$^2$ during the year following bariatric surgery.

Cross-sectional data support these results. *Pramudji et al. (2019)* reported that CRP concentrations increased with the number of C-alleles of the IL-6 rs1800795 polymorphism for obese ($P = 0.02$) but not non-obese Indonesians ($P = 0.64$), consistent with the higher average CRP concentrations of those who were obese (2.26 vs. 0.49 mg/L, Fig. 4A). *Teng et al. (2009)* reported that the effects of obesity on Taiwanese CRP concentrations differed significantly by rs2794521 ($P_{interaction} = 0.03$, Fig. 4B histogram) and rs1800947 ($P_{interaction} = 0.02$, Fig. 4C histogram), and possibly rs1205 (Fig. 4D histogram). Correspondingly, average CRP levels were approximately twice as high in the obese than non-obese subjects, and as shown in the associated line graphs, the interactions could be attributed to a larger genetic effect size at higher average CRP concentrations. Studies by *Eiriksdottir et al. (2009)*, *Todendi et al. (2016)*, and *Wang et al. (2011)* all present results consistent with a larger rs1205 genotype differences at the higher average CRP concentrations of those who are more overweight. *Eiriksdottir et al. (2009)* reported that the log CRP difference between rs1205 G-carriers and AA homozygotes increased as CRP levels increased with increasing BMI in both men ($P_{interaction} = 0.05$) and women ($P_{interaction} = 0.09$).

## Smoking

The Speedwell Survey of British men reported that average CRP increased significantly from those who never smoked (1.13 mg/L), to those who averaged 1–14 (1.87 mg/L), 15–24 (2.32 mg/L), and greater than 25 cigarettes/day (2.05 mg/L) (*Lowe et al., 2001*). Consistent with quantile-dependent expressivity, *Friedlander et al. (2006)* reported that the heritability of untransformed CRP was 4-fold larger in smokers than nonsmokers

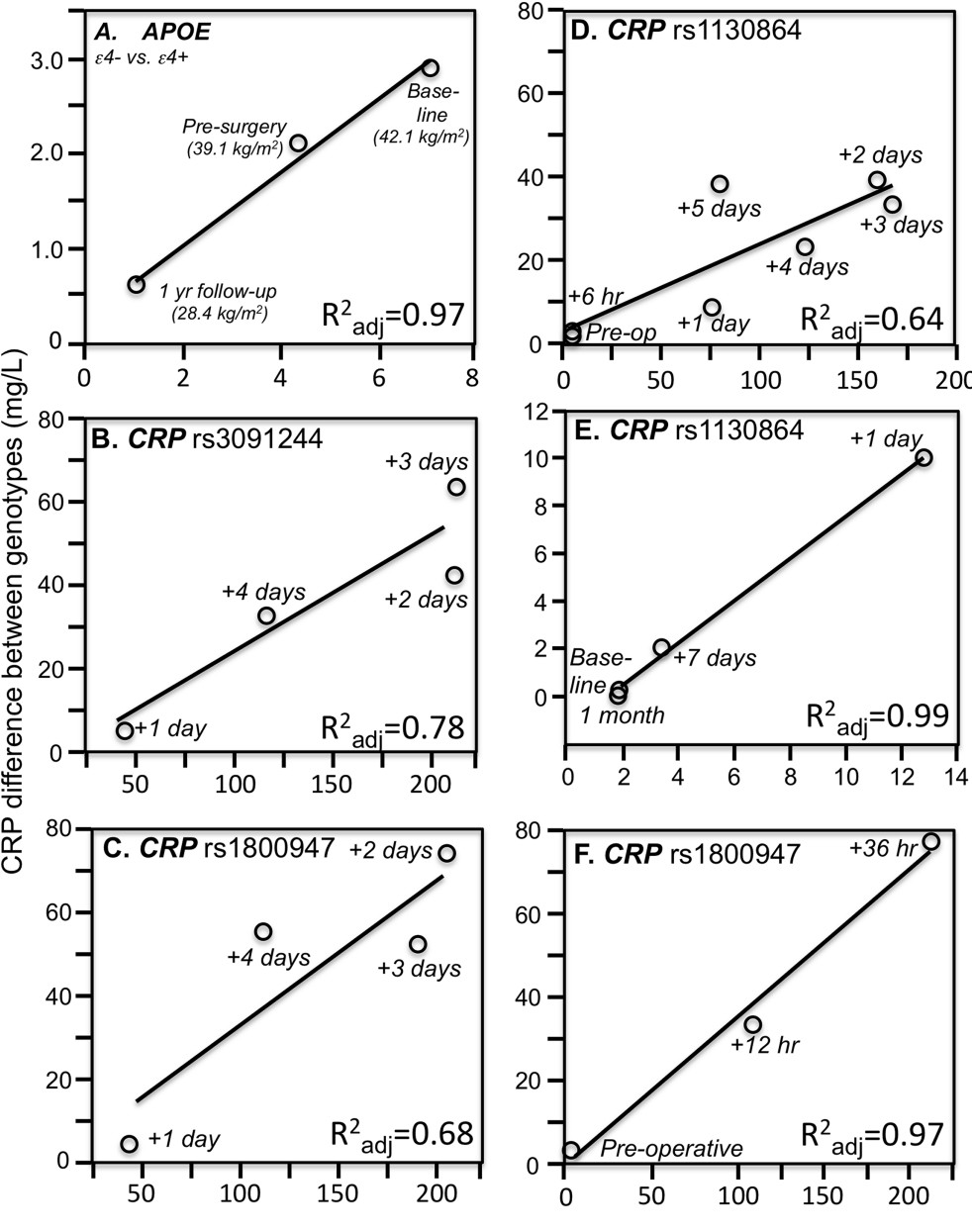

**Figure 3 Simple regression analysis showing larger genotype differences associated with higher estimated average CRP response.** Simple regression analysis of showing larger genotype differences associated with higher estimated average CRP response for the data presented in: (A) *Farup, Rootwelt & Hestad (2020)* report on the *APOE* CRP differences (non-carriers minus carriers of ε4-allele) in morbidly obese patients losing weight; (B) *Perry et al. (2009)* report on the rs3091244 CRP difference (T-allele carrier minus noncarrier) post CABG surgery ($P_{linear}$ = 0.08); (C) *Perry et al. (2009)* report on the rs1800947 CRP difference (GG homozygotes minus C-allele carrier) post CABG surgery ($P_{linear}$ = 0.11); (D) *Brull et al. (2003)* report on the rs1130864 CRP difference (TT homozygotes minus C-allele carriers) pre- and post CABG surgery ($P_{linear}$ = 0.02); (E) *D'Aiuto et al. (2005)* report on the rs1130864 CRP difference (TT homozygotes minus C-allele carriers) following periodontal intensive therapy ($P_{linear}$ = 0.002); (F) *Motoyama et al. (2009)* report on the rs1800947 CRP difference (GG homozygotes minus C-allele carriers) following esophagectomy surgery ($P_{linear}$ = 0.07).

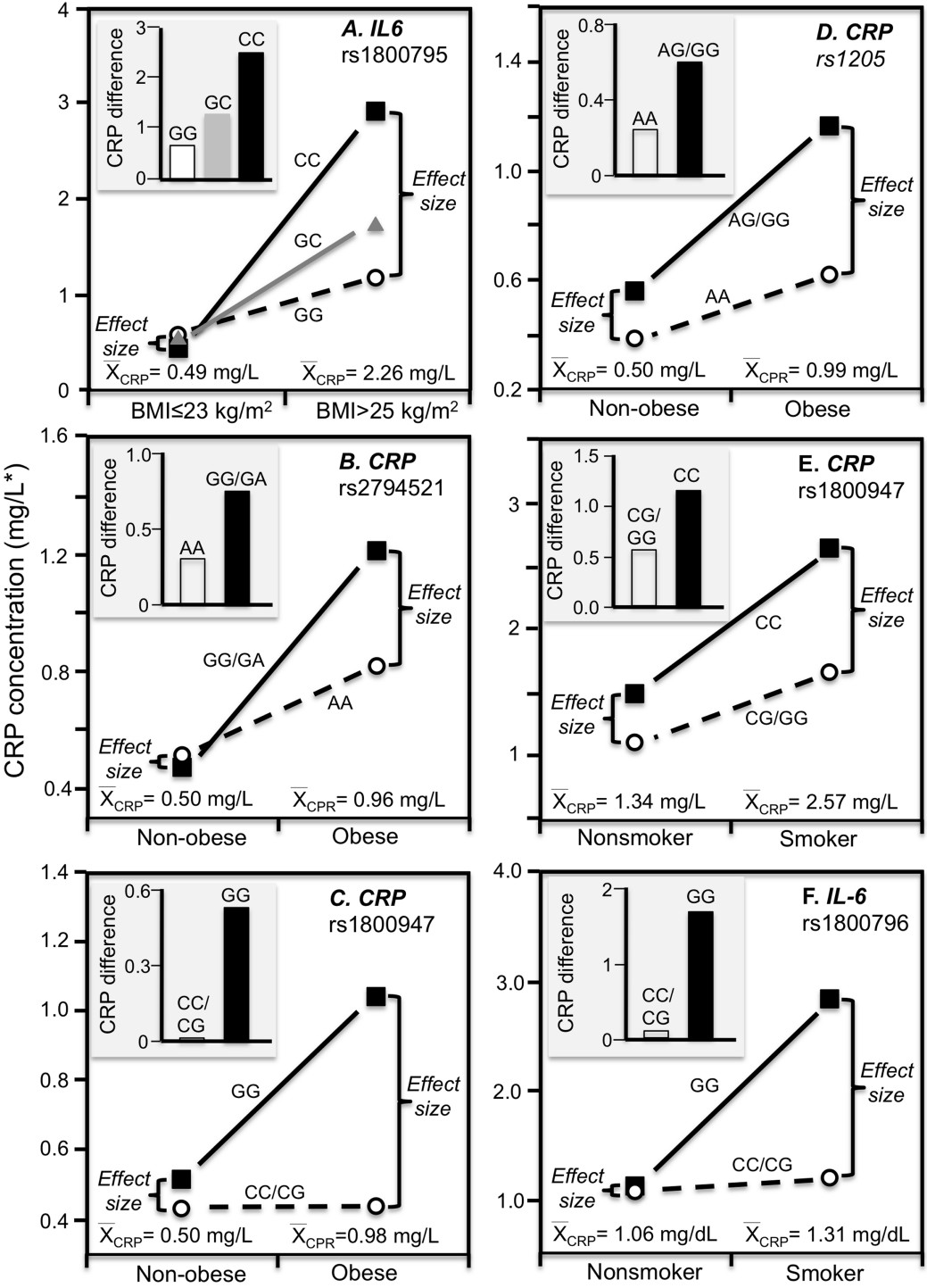

**Figure 4 Precision medicine perspective of genotype-specific CRP differences (histogram inserts) vs. quantile-dependent expressivity perspective (line graphs).** Precision medicine perspective of genotype-specific CRP differences (histogram inserts) vs. quantile-dependent expressivity perspective (line graphs showing larger genetic effect size when average CRP concentrations were high) for the data presented in: (A) *Pramudji et al. (2019)* of the CRP difference between obese and non-obese subjects by the -174 G>C *IL-6* polymorphism; (B) *Teng et al. (2009)* report on the CRP difference between obese and non-obese subjects by the rs2794521 genotypes ($P_{interaction}$ = 0.034); (C) *Teng et al. (2009)* report on the CRP difference between obese and non-obese subjects by rs1800947 genotypes ($P_{interaction}$ = 0.02);

**Figure 4** (continued)
(D) *Teng et al. (2009)* report on the CRP difference between obese and non-obese subjects by the rs1205 genotypes ($P_{interaction}$ = 0.02); (E) *Luetragoon et al. (2018)* report on the CRP difference between smokers and nonsmokers by *CRP* 1800947 genotypes; (F) *Shin et al. (2007)* report on the CRP difference (mg/dL) between smokers and nonsmokers by *IL6* rs1800796 genotypes. *Except where noted.

(0.863 vs. 0.193), and *Retterstol, Eikvar & Berg (2003)* reported a higher MZ twin correlation in smokers than nonsmokers ($r$ = 0.49 vs. $r$ = 0.34).

   *Luetragoon et al. (2018)* reported a significant CRP difference between smokers and nonsmokers in the *CRP* rs1800947 CC homozygotes ($P$ = 0.03) but not G-allele carriers ($P$ = 0.67, Fig. 4E histogram), corresponding to a larger genotype difference for the higher CRP concentrations of the smokers vs. nonsmokers (2.57 vs. 1.34 mg/L, $P$ = 0.009 line graph). *Shin et al. (2007)* reported a significant interaction between smoking and the IL-6 rs1800796 promoter polymorphism in their effect on CRP concentrations ($P_{interaction}$ = 0.04). Whereas the Fig. 4F histogram shows that the effect of smoking on CRP was greater in GG homozygotes, the line graph suggests that the results could also be interpreted in part as a larger genetic effect size at the higher CRP concentrations of the smokers. Data presented by *Gander et al. (2004)* in their figure 1 suggest a greater smoking effect in carriers of the A-allele than GG homozygotes of the TNF-α rs1800629 polymorphism (Fig. 5A), corresponding to a larger difference between genotypes at the higher average estimated concentrations of the smokers.

## Physical activity

*Brull et al. (2003)* reported overall mean CRP concentrations in British army recruits increased significantly following an intensive 48-h final military endurance exercise (1.14 mg/L post-exercise vs. 0.59 mg/L at baseline). Figure 5B shows that the exercise-induced CRP increases were over 2.5-fold greater in rs1130864 TT homozygotes than C-allele carriers (histogram), and that the difference between genotypes was two-fold greater 2 h post exercise than at baseline (1.28 vs. 0.49 mg/L difference) corresponding to the higher post-exercise mean concentrations (line graph).

## Diet

A Mediterranean-style diet that is rich in monounsaturated fat, polyunsaturated fat, and fiber was reported to significantly decrease CRP concentrations relative to a prudent diet (*Esposito et al., 2004*). In T2DM, *Keramat et al. (2017)* reported a greater effect of monounsaturated fat intake on CRP concentrations in CC homozygotes of the APOA2 rs5082 polymorphism than in carriers of the T-allele (Fig. 5C histogram, $P_{interaction}$ = 0.02). The line graph suggests there were greater genotype differences and average CRP concentrations below vs. above median intake of monounsaturated fatty acids.

   Our analysis of *Carvalho-Wells et al. (2012)* data suggest that *APOE* ε3ε3 subjects who switched from an 8-week low fat to an 8-week high fat diet had somewhat greater increases in CRP than ε3ε4 subjects. Figure 5D shows a larger difference between genotypes at the significantly higher CRP concentrations of the high-fat vis-à-vis the

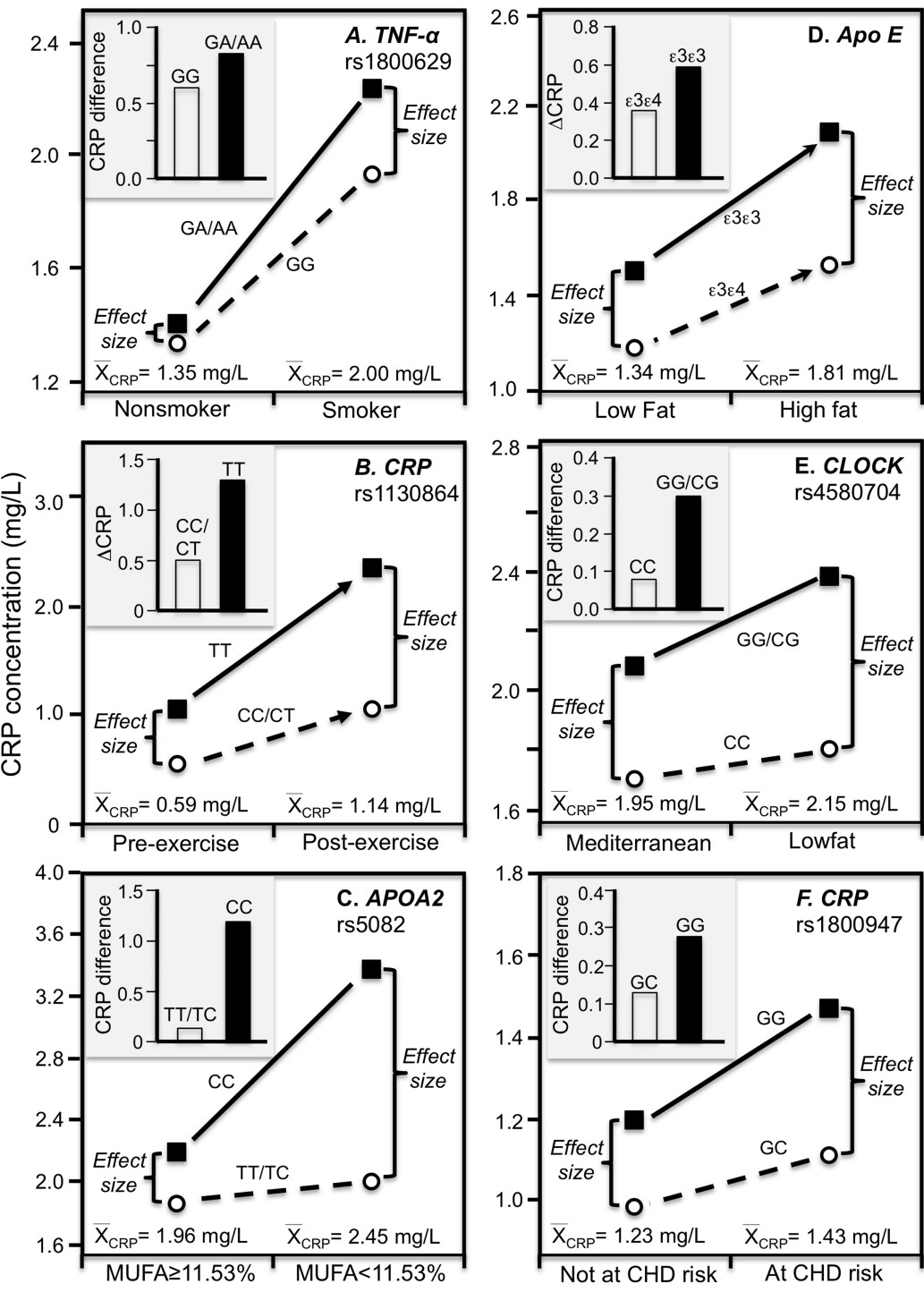

**Figure 5  Precision medicine perspective of genotype-specific CRP differences (histogram inserts) vs. quantile-dependent expressivity perspective (line graphs).** Precision medicine perspective of genotype-specific CRP differences (histogram inserts) vs. quantile-dependent expressivity perspective (line graphs showing larger genetic effect size when average CRP concentrations were high) for the data presented in: (A) *Gander et al. (2004)* report on the CRP difference between smokers and nonsmokers by *CRP* rs1800629; (B) *Brull et al. (2003)* reported on the effect of 48-h military endurance exercise on CRP concentrations by rs1130864 genotypes; (C) *Keramat et al. (2017)* report on the effect of monounsaturated fat intake on CRP concentrations by *APOA2* rs5082 genotypes;

**Figure 5** (continued)
(D) *Carvalho-Wells et al. (2012)* report on the effect of a high fat diet by *APOE* isoform; (E) *Gomez-Delgado et al. (2015)* report on the effect of a lowfat diet by *CLOCK* rs4580704 genotypes; (F) *Zee & Ridker (2002)* report on the CRP difference between men experiencing vs. not experiencing their first arterial thrombosis during 8.6 year follow-up by *CRP* rs1800947 genotypes.

low-fat diet. Supplementing the high-fat diet with 3.45 g of DHA-rich oil eliminated the genotype difference (not displayed).

*Gomez-Delgado et al. (2015)* reported that decreases in CRP concentrations from switching from the basal to a low-fat diet were greater in CC-homozygotes of the circadian locomotor output cycles kaput (CLOCK) rs4580704 polymorphism than in carriers of the G-allele ($P < 0.001$). Cross-sectionally, the histogram of Fig. 5E shows that the CRP difference between consuming a low fat diet vs. a Mediterranean diet for 1 year was greater for carriers of the G-allele than CC homozygotes (histogram), while the line graph shows that the difference between rs4580704 genotypes was greater for the higher CRP concentrations of the low-fat diet than for the lower CRP concentrations of the Mediterranean diet.

### Elevated coronary heart disease risk

*Zee & Ridker (2002)* reported that baseline median CRP concentrations in healthy men who experienced their first arterial thrombosis (nonfatal MI, nonfatal stroke, or cardiovascular death) during 8.6-year follow-up were significantly higher than matched controls who remained event free (1.43 vs. 1.23 mg/L, $P = 0.006$). The CRP difference between those experiencing and not experiencing thrombosis was greater in rs1800947 GG homozygotes than GC heterozygotes (Fig. 5F histogram), which corresponded to a larger genotype difference at the higher median baseline CRP concentrations of those with a thrombotic destiny (Fig. 5F line graph).

### Myocardial infarction survivors

Data reported by *Kovacs et al. (2005)* showed that the CRP difference between carriers and non-carriers of the rs3091244 A-allele was greater in myocardial infarction survivors ($P < 0.02$) than matched controls (NS), consistent with the higher estimated median concentrations of the survivors (1.46 vs. 0.96 mg/L, Fig. 6A).

### Stroke

*Ben-Assayag et al. (2007)* reported that CRP differences between G-allele carriers and AA homozygotes of the rs2794521 polymorphism were significant at admission following a stroke or transient ischemic attack (2.02 vs. 1.73 mg/L, $P = 0.03$) when average CRP concentrations were elevated (1.71 g/L) but not 6-months later (1.44 vs. 1.43 mg/L, $P = 0.98$) when average CRP concentrations were lower (1.43 mg/L). The CRP change between admission and 6-month follow-up was significantly greater in the G-allele carriers than AA homozygotes ($P = 0.05$).

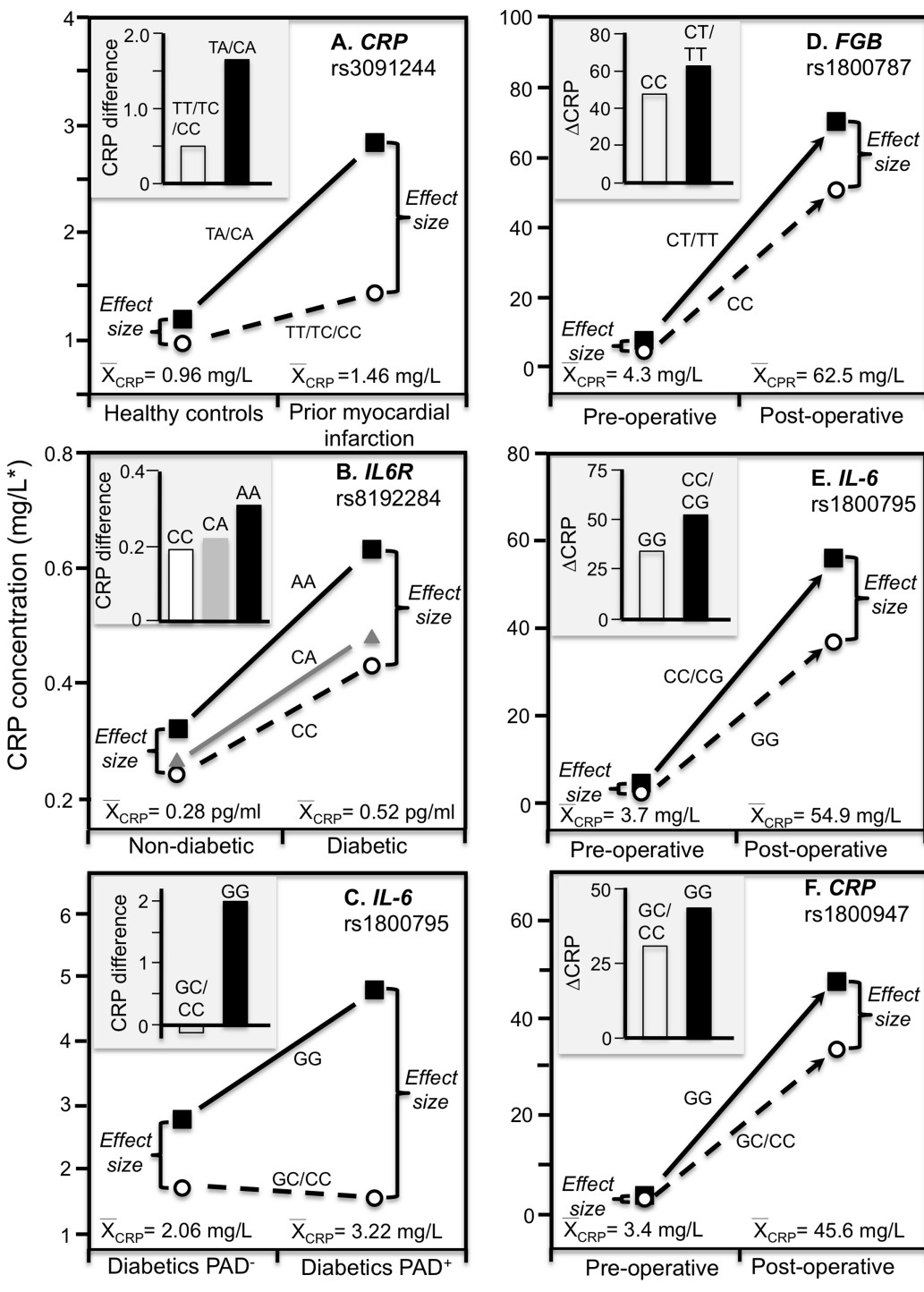

**Figure 6 Precision medicine perspective of genotype-specific CRP differences (histogram inserts) vs. quantile-dependent expressivity perspective (line graphs).** Precision medicine perspective of genotype-specific CRP differences (histogram inserts) vs. quantile-dependent expressivity perspective (line graphs showing larger genetic effect size when average CRP concentrations were high) for the data presented in: (A) *Kovacs et al. (2005)* reported on the effect of myocardial infarction by *CRP* rs3091244 genotypes cross-sectionally; (B) *Qi, Rifai & Hu (2007)* report on the effect of T2DM by interleukin-6 receptor (*IL6R*) rs8192284 genotypes ($P_{interaction}$ = 0.03); (C) *Libra et al. (2006)* report on the CRP difference between T2DM with (PAD+) and without (PAD−) peripheral arterial disease by *IL-6* G(-174)C

**Figure 6** (continued)
rs1800795 genotypes; (D) *Wypasek et al. (2012)* 2012 reported on the effects of coronary artery bypass grafting (CABG) surgery on CRP by fibrinogen beta-chain (*FGB*) -148C>T genotypes (rs1800787); (E) *Wypasek et al. (2010)* reported on the effects of CABG surgery on CRP by -174G>C *IL-6* (rs1800795) genotypes; (F) *Mathew et al. (2007)* report on the effects of CABG with cardiopulmonary bypass by *CRP* +1059G>C (rs1800947) genotypes. *Except where noted.

### Type 2 diabetes mellitus

Subclinical systemic inflammation contributes to the etiology of insulin resistance (*Kolb, 2005*), which may explain the increased diabetes risk associated with elevated CRP concentrations prospectively (*Pradhan et al., 2001*). The missense variant rs8192284 of the interleukin-6 receptor (IL6R) gene is reported to be strongly associated with IL-6 and CRP concentrations in genomewide association studies (*Qi, Rifai & Hu, 2007*). *Qi, Rifai & Hu (2007)* reported a significant interaction ($P_{interaction}$ = 0.03) between diabetes and rs8192284 in their effect on CRP concentrations (Fig. 6B histogram). However, diabetic had higher estimated CRP concentrations than non-diabetics, and the interaction could be due to the larger genetic effect size at the higher average CRP concentrations of the T2DM (Fig. 6B line graph).

### Peripheral arterial disease

The IL-6 rs1800795 polymorphism has been suggested to affect IL-6 expression and influence the development of PAD, a vascular pathology associated with T2DM (*Libra et al., 2006*). Data reported by *Libra et al. (2006)* showed that the CRP difference between T2DM patients with and without PAD was greater in GG homozygotes than C-allele carriers (2.0 ± 0.34 vs. −0.16 ± 0.26 mg/L, Fig. 6C histogram). Average CRP concentrations were higher in the PAD+ than PAD- patients (3.22 ± 0.16 vs. 2.06 ± 0.13 mg/L), and correspondingly, the difference between genotypes was greater for PAD+ than PAD- (3.25 ± 0.32 vs. 1.09 ± 0.28 mg/L).

### Sex

Females have higher CRP concentrations than men, which may be hormonal, that is, female CRP concentrations correlate positively with estradiol levels, and the odds of CRP falling above the median doubles with each standard deviation increment in endogenous estradiol (*Eldridge et al., 2020*). Higher female CRP may explain the greater estimated heritability we observed in female than male offspring (0.13 vs. 0.08) and female than male sibling (0.20 vs. 0.10), the greater heritability of untransformed CRP in females than males reported by *Friedlander et al. (2006)* (0.352 vs. 0.150), and the higher within-pair correlations in female than male MZ twins reported by *Retterstol, Eikvar & Berg (2003)* ($r$ = 0.44 vs. $r$ = 0.31).

### Race

CRP concentrations tend to be higher in Blacks than other racial groups (*Nazmi & Victora, 2007*), that is, mean CRP concentrations estimated from meta-analysis are 2.6 mg/L for

African–Americans, 2.51 for Hispanics, 2.03 for White Americans, and 1.01 for East Indians (*Shah et al., 2010*). European ancestry is negatively correlated with age-adjusted CRP in both African–Americans ($P < 0.0001$) and Hispanic Americans ($P = 0.001$) (*Reiner et al., 2012*). Quantile-dependent expressivity may contribute to the higher heritability of lnCRP in Blacks than whites (53% vs. 31%) reported by *Wu et al. (2009)*.

### Acute phase response

Rapid hepatic synthesis of CRP occurs as part of the acute phase response to infection, injury or trauma (*Pepys, 2003*; *Agrawal, 2005*). The increase can be 1,000-fold (*Pepys, 2003*; *Agrawal, 2005*). Consistent with quantile-dependent expressivity, several SNPs show effects on CRP that are greatly accentuated during acute phase response vis-à-vis their basal concentrations, and intermediate effects during intermediate transitional concentrations.

Coronary artery bypass grafting (CABG) surgery produces a strong inflammatory response with substantially increased CRP, fibrinogen, and IL-6 circulating concentrations (*Brull et al., 2003*; *Wypasek et al., 2012*). *Wypasek et al. (2012)* reported that CRP increased from a pre-operative concentration of $4.3 \pm 0.1$ mg/L to $62.5 \pm 4.2$ mg/L 5–7 days following CABG surgery ($P < 0.0001$). Consistent with quantile-dependent expressivity, the line graph of Fig. 6D shows that the increase in mean concentrations coincided with substantially greater post-operative CRP differences between carriers and non-carriers of the T allele of the fibrinogen beta-chain (*FGB*) −148C>T rs1800787 polymorphism ($70.4 \pm 5.0$ vs. $51.6 \pm 4.25$ mg/L, $P = 0.005$) vis-à-vis their much smaller pre-operative difference ($7.49 \pm 1.2$ vs. $4.26 \pm 1.0$ mg/L, $P = 0.04$). Another report by *Wypasek et al. (2010)* showed that post-operative CRP concentrations were significantly higher in C-allele carriers than non-carriers of the IL-6 rs1800795 polymorphism ($56.39 \pm 4.27$ vs. $36.60 \pm 7.78$ mg/L, $P = 0.03$) when average CRP concentrations were elevated ($54.9 \pm 3.8$ mg/L), which was substantially greater than the pre-operative difference between genotypes ($4.1 \pm 0.35$ vs. $2.4 \pm 0.59$ mg/L, $P = 0.02$) when average CRP concentrations were much lower ($3.71 \pm 0.45$, Fig. 6E).

*Mathew et al. (2007)* reported a substantial increase in mean CRP concentrations following CABG and cardiopulmonary bypass that was significantly affected by the rs1800947 polymorphism (Fig. 6F, $P = 0.01$). Twenty-four hour post cross-clamp CRP levels were significantly higher in GG homozygotes than CC homozygotes and CG heterozygotes ($P < 0.001$). The greater post-operative CRP increase in GG than C-allele carriers (histogram) corresponds to a small pre-operative genotype difference when the average CRP concentration was 3.4 mg/L vs. a large postoperative genotype difference when the average CRP concentration was 45.6 mg/L.

*Perry et al. (2009)* reported that median peak CRP went from 1.2 mg/L preoperatively to 293.3 mg/L postoperatively following CABG surgery. The rs3091244 T-allele was associated with higher peak postoperative CRP ($P = 2.1 \times 10^{-3}$), whilst the rs1800947 C-allele of was associated with lower peak postoperative levels ($P = 2.4 \times 10^{-4}$). Compared to their most common haplotype (rs1800947G/rs3091244C), the peak postoperative levels

were significantly lower for haplotype 4 (CC, $P = 0.004$) and significantly higher for haplotype 2 (GT, $P = 0.03$). Figures 3B and 3C show that the postoperative genotype differences increased with increasing CRP concentrations, consistent with quantile-dependent expressivity.

Brull et al. (2003) reported an 83-fold increase in average CRP, from a preoperative average of $1.97 \pm 0.36$ mg/L to a post-operative average of $167.2 \pm 5.0$ mg/L 72 h after CABG surgery ($P < 0.0005$), and that CRP concentrations remained significantly elevated through post-operative day five ($P < 0.0005$). The rs1130864 TT-homozygotes had significantly higher CRP levels than C-allele carriers at all time points >24 h post-operation, but not before. Our analysis of their figure 2A suggests that rs1130864 genotype differences were significantly related to average CRP concentrations during the acute phase response (Fig. 3D, $P = 0.02$).

Intensive periodontal therapy also causes sharp rises in CRP and IL-6 that peak by 24 h and remain elevated for up to 7 days (D'Aiuto et al., 2004). D'Aiuto et al. (2005) reported significantly higher CRP concentrations in rs1130864 TT homozygotes than C-allele carriers one (21.10 vs. 12.37 mg/L, $P = 0.02$) and 7-days (4.89 vs. 3.08 mg/L, $P < 0.01$) during the inflammatory stimulus of periodontal intensive therapy. Correspondingly, the geometric means of CRP concentrations were elevated one (13.64 mg/L, $P < 0.0001$) and 7 days (3.35 mg/L, $P < 0.0001$) relative to baseline (1.93 mg/L), such that the intermediate 7-day genotype difference was as predicted by linear interpolation using the 7-day average CRP concentration relative to baseline and day one average concentrations (Fig. 3E). Similarly, Motoyama et al. (2009) data showed that the CRP difference between rs1800947 GG homozygotes than C-allele carriers after curative esophagectomy was linearly related to average CRP concentrations, and that the intermediate 12 h genotype difference was almost exactly predicted by it's intermediate average concentration by linear interpolation (Fig. 3F).

C-reactive protein concentrations also increase substantially during acute ischemia and return to near basal levels during the chronic stable phase after ischemia is resolved (Suk Danik et al., 2006). Recurrent myocardial infarction and cardiovascular death are strongly related to CRP increases during acute coronary syndrome (Suk Danik et al., 2006). Suk Danik et al. (2006) reported that rs3091244 AA homozygotes had the highest (76.6 mg/L) median concentrations during the acute rise in plasma CRP-concentrations following an acute coronary syndrome whereas the median concentration in noncarriers was 11.1 mg/L. Figures 7A–7C show that during both acute coronary syndrome and the chronic stable phase 1 month later, CRP concentrations were significantly higher in rs3091244 A-allele carriers than non-carriers ($P = 0.0005$ and $P = 0.0008$, respectively), rs1800947 GG-homozygotes than C-allele carriers (both $P < 0.0001$), and per dose of the rs1205 G-allele (both $P < 0.0001$). Consistent with quantile-dependent expressivity, the line graphs show greater genotype differences during acute coronary syndrome when median CRP concentrations were substantially elevated vis-a-vis the chronic stable phase. Results reported by Kovacs et al. (2005) for rs3091244 are consistent with Suk Danil's results (Fig. 7D).

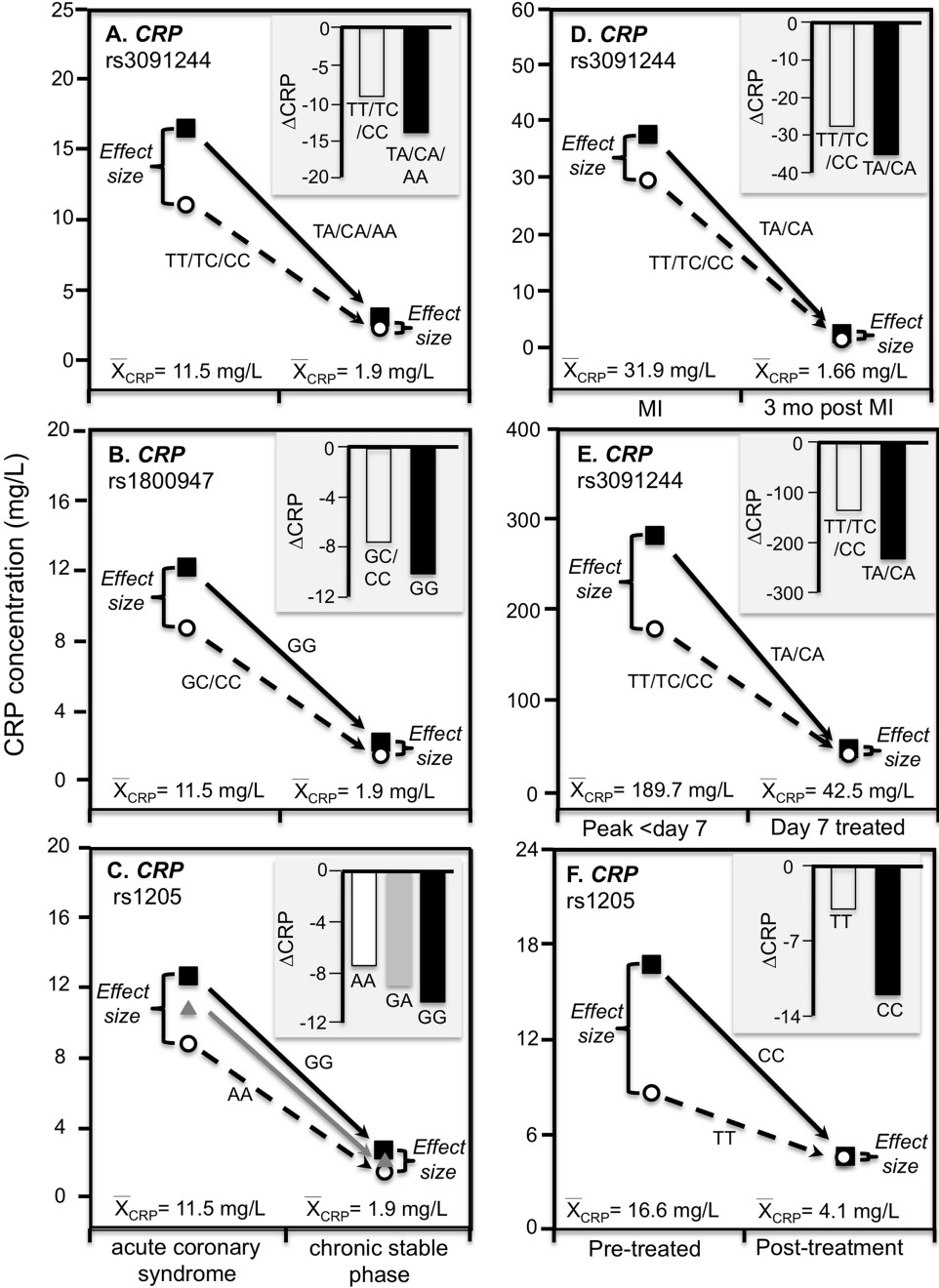

**Figure 7 Precision medicine perspective of genotype-specific CRP differences (histogram inserts) vs. quantile-dependent expressivity perspective (line graphs).** Precision medicine perspective of genotype-specific CRP differences (histogram inserts) vs. quantile-dependent expressivity perspective (line graphs showing larger genetic effect size when average CRP concentrations were high) for the data presented in: (A) *Suk Danik et al. (2006)* report on the effect of acute coronary syndrome by *CRP* rs3091244 genotypes; (B) *Suk Danik et al. (2006)* report on the effect of acute coronary syndrome by *CRP* rs1800947 genotypes; and (C) *Suk Danik et al. (2006)* report on the effect of acute coronary syndrome by *CRP* rs1205 genotypes; (D) *Kovacs et al. (2005)* report on the effect of myocardial infarction (MI) by *CRP* rs3091244 genotypes longitudinally; (E) *Mölkänen et al. (2010)* reported on the effect of *Staphylococcus aureus* bacteremia by rs3091244 genotypes; (F) *Ammitzbøll et al. (2014)* report on the effect of early chronic rheumatoid arthritis by *CRP* rs1205.  

### Infection

*Mölkänen et al. (2010)* reported greater differences in CRP concentrations between carriers and non-carriers of the rs3091244 A-allele at peak CRP concentrations (103 mg/L difference, $P = 0.004$) during the first week of a *Staphylococcus aureus* bacteremia when average CRP was approximately 190 mg/L, than 7-days after diagnosis (5 mg/L difference, $P = 0.77$) when average CRP concentrations had decreased to approximately 43 mg/L (Fig. 7E).

### Chronic rheumatoid arthritis

*Ammitzbøll et al. (2014)* reported that rs1205 TT homozygotes had 50% lower CRP concentrations than CC homozygotes at baseline ($P = 0.005$) when average concentrations were approximately 16.6 mg/L in patients with untreated early chronic rheumatoid arthritis, but not after 1-year ($P = 0.38$) when antirheumatic drug and steroid treatment had decreased average CRP concentrations to approximately 4.1 mg/L (Fig. 7F). Another study of rheumatoid arthritis patients by *Rhodes et al. (2010)* compared CRP concentrations across genotypes using erythrocyte sedimentation rate (ESR) as an independent measure of inflammation. Their data showed larger estimated CRP differences between genotypes at an ESR of 80 vs. 40 for rs1800947 (CC/GC/GG: 19.4/28.6/42.2 vs. 12.0/17.7/26.1 mg/L), rs1205 (AA/GA/GG: 27.6/35.5/45.7 vs. 17.0/21.9/28.2 mg/L), and rs11265257 (AA/GA/GG: 29.2/37.3/47.6 vs. 17.9/22.9/29.2 mg/L), which quantile-dependent expressivity would attribute to the higher average CRP when ESR was 80 than 40 (approximately 40 vs. 24 mg/L). *Wielińska et al. (2020)* reported larger differences between genotypes for genes coding for the receptor activator of nuclear factor κB (RANK rs8086340) and its ligand (RANKL rs7325635) in rheumatoid arthritis patients prior to 12 weeks of anti-TNF treatment when average CRP concentrations were high (23.6 mg/L), than after treatment when average concentrations were lower (9.84 mg/L, Figs. 8A and 8B).

### Inflammatory bowel disease

TNF-α is both a major regulator of hepatic CRP production and a key inflammatory mediator in IBD pathophysiology (*Hampe et al., 1999*). Data presented by *Vatay et al. (2003)* show median CRP concentrations were substantially higher in GA heterozygotes than GG homozygotes of the TNF-α rs1800629 polymorphism for the high CRP concentrations of active phase IBD, but not for the low CRP concentrations of matched healthy controls (Fig. 8C).

### Ankylosing spondylitis

This is a spinal inflammation whose severity, clinical progression, and treatment response are indicated by elevated CRP concentrations. Etanercept, a TNF-α inhibitor, is one of the few treatment options for ankylosing spondylitis. *Xu, Jiang & Zhang (2020)* reported that rs3091244 AA homozygotes have higher CRP concentrations than carriers of the G allele both before and after 12-week etanercept treatment, but that this difference in genotypes was over two-fold greater prior to treatment when average CRP was high vis-à-vis post-treatment concentrations (Fig. 8D).

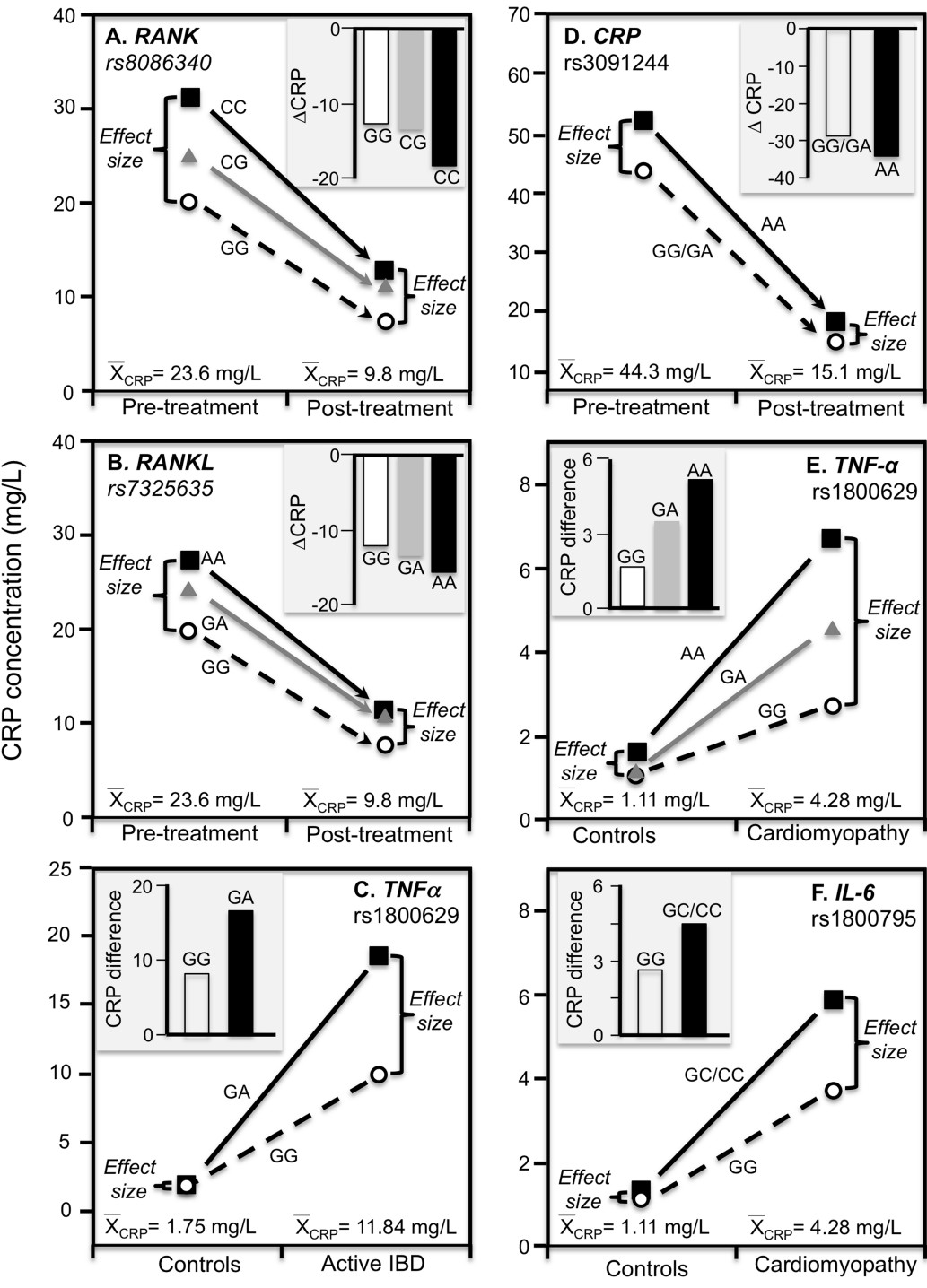

**Figure 8 Precision medicine perspective of genotype-specific CRP differences (histogram inserts) vs. quantile-dependent expressivity perspective (line graphs).** Precision medicine perspective of genotype-specific CRP differences (histogram inserts) vs. quantile-dependent expressivity perspective (line graphs showing larger genetic effect size when average CRP concentrations were high) for the data presented in: (A) *Wielińska et al. (2020)* report on the effect of anti-TNF treatment by *RANK* rs8086340 genotypes; (B) *Wielińska et al. (2020)* report on the effect of anti-TNF treatment by *RANKL* rs7325635 genotypes; (C) *Vatay et al. (2003)* report on the CRP difference between active phase inflammatory bowel disease and healthy controls by tumor necrosis factor alpha (*TNF-α*) G-308A (rs1800629)

**Figure 8 (continued)**
promoter polymorphism; (D) *Xu, Jiang & Zhang (2020)* report on the effect of etanercept treatment in Ankylosing spondylitis patients by *CRP* rs3091244 genotypes; (E) *Liaquat et al. (2014)* report on the effect of idiopathic dilated cardiomyopathy by *TNF-α* (rs1800629) -308G>A genotypes; (F) *Liaquat et al. (2014)* report on the effect of idiopathic dilated cardiomyopathy by *IL-6* rs1800795 (−174 G>C) genotypes.

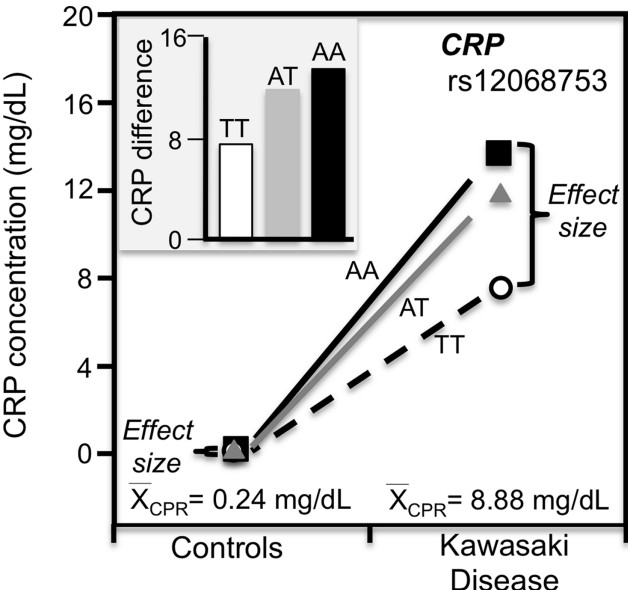

**Figure 9 Precision medicine perspective of genotype-specific CRP differences (histogram inserts) vs. quantile-dependent expressivity perspective (line graph).** Precision medicine perspective of genotype-specific CRP differences (histogram inserts) vs. quantile-dependent expressivity perspective (line graphs showing larger genetic effect size when average CRP concentrations were high) for the data presented in *Kim et al. (2015)* report on the effect of Kawasaki disease by *CRP* promoter rs12068753 genotypes.

### Dilated cardiomyopathy

Proinflammatory cytokines may contribute to dilated cardiomyopathy, a condition distinguished by dilatation and impaired contraction of the left or both ventricles. *Liaquat et al. (2014)* reported that differences in CRP concentrations between idiopathic dilated cardiomyopathy patients and healthy controls increased with the number of A-alleles of the TNF-α rs1800629 polymorphism (Fig. 8E histogram), and were greater in C-allele carriers of the IL-6 rs1800795 polymorphism (Fig. 8F histogram). Consistent with quantile-dependent expressivity, the line graphs show that the effects of the genotypes were greater for the higher mean concentrations of the patients than controls.

### Kawasaki disease

Kawasaki disease is an inflammation of the walls of medium-size arteries that primarily affect children. *Kim et al. (2015)* reported that the CRP promoter rs12068753 showed greater CRP differences between genotypes in patients with Kawasaki disease than controls in accordance with the cases' higher average CRP concentrations (8.9 vs. 0.3 mg/dL, Fig. 9).

*Exceptions*

Contrary to expectations: (1) *Wu et al. (2011)* reported that the significant interaction between activating transcription factor (*AFT3*) rs10475 and obesity on CRP concentrations ($P_{\text{interaction}}$ = 0.006) was due to a significant difference between genotypes ($P$ = 0.001) in non-obese subjects having lower overall CRP concentrations and not obese subjects ($P$ = 0.27) whose CRP concentrations were higher; (2) *Keramat et al. (2017)* reported significantly greater *APOA2* rs5082 genotype differences for the lower average CRP concentrations of low saturated fat intake than for the higher average CRP concentrations above median saturated fat intake: (3) *Hsu et al. (2011)* report of significantly greater genotype differences for hepatic nuclear factor-1α (*HNF1A*) rs1920792, rs2464196, and rs1169310 polymorphisms in nonobese than obese subjects despite the higher average CRP the obese; (4) *Eklund et al. (2006)* report that CRP differed between *IL6* rs1800795 genotypes after weight loss when average CRP concentrations were decreased but not before when average concentrations were higher; (F) *Retterstol, Eikvar & Berg (2003)* report of a larger MZ correlation below the median BMI than above ($r_{\text{MZ}}$ = 0.42 vs. 0.31) despite the positive correlation between BMI and CRP. These exceptions to quantile-dependent expressivity may make them noteworthy in themselves, however, most reported gene-environment interactions are unreplicated, and it is expected that at least some of the reported interactions could be spurious.

## CONCLUSION

Heritability of serum CRP concentration is quantile-specific, which may explain or contribute to the inflated CRP differences between *CRP* (rs1130864, rs1205, rs1800947, rs2794521 rs3091244), *FGB* (rs1800787), *IL-6* (rs1800795, rs1800796), *IL6R* (rs8192284), *TNF*-α (rs1800629) and *APOE* genotypes following CABG surgery, stroke, TIA, curative esophagectomy, intensive periodontal therapy, or acute exercise; during acute coronary syndrome or *Staphylococcus aureus* bacteremia; or in patients with chronic rheumatoid arthritis, diabetes, peripheral arterial disease, ankylosing spondylitis, obesity or inflammatory bowel disease or who smoke.

Quantile-dependent expressivity is a novel concept, and unsurprisingly, the majority of articles do not provide the data in a form necessary to evaluate its applicability, namely genotype-specific CRP concentrations stratified by characteristics affecting average CRP concentrations. Although it is reported that CRP concentrations are higher in patients with abdominal aortic aneurysm (*Shangwei et al., 2017*), poor cognitive performance and cognitive decline over time (*Yaffe et al., 2003*), anxiety disorders (*Naudé et al., 2018*), and Alzheimer's disease (*Zaciragic et al., 2007*), it is not known whether these conditions affect the effect size of CRP-related genetic variants.

Finally, we note that quantile regression and its bootstrap-derived standard errors do not require a normal distribution, and provide insights into CRP inheritance heretofore unstudied. The decision to logarithmically transform CRP concentration has been exclusively based on the theoretical requirement of the parametric statistical testing rather than a biological rationale. All the major genomewide association studies were performed on log CRP, as were virtually all tests of association or gene-environment interaction.

This statistical accommodation may work against the goal of identifying some SNPs affecting CRP concentrations given our results suggesting the largest genetic effects are at the highest concentrations.

## ABBREVIATION KEY

| | |
|---|---|
| AFT3 | Activating transcription factor |
| APOA2 | Apolipoprotein A2 |
| APOE | Apolipoprotein E |
| $\beta_{FS}$ | Full-sib regression slope |
| $\beta_{OP}$ | Offspring-parent regression slope |
| BMI | Body mass index |
| CABG | Coronary Artery Bypass surgery |
| CHD | Coronary Heart Disease |
| CLOCK | Circadian locomotor output cycles kaput |
| CRP | C-reactive protein |
| ESR | Erythrocyte sedimentation rate |
| FGB | Fibrinogen beta-chain |
| $h^2$ | Heritability in the narrow sense |
| HNF1A | hepatic nuclear factor-1α |
| IL-6 | Interleukin-6 |
| IL6R | Interleukin-6 receptor |
| IBD | Inflammatory bowel disease |
| MI | Myocardial infarction |
| NHLBI | National Heart Lung and Blood Institute |
| NS | Not statistically significant (P>0.05) |
| PAD | Peripheral arterial disease |
| RANK | Receptor activator of nuclear factor κB |
| RANKL | Receptor activator of nuclear factor κB ligand |
| SD | Standard deviation |
| SE | Standard error |
| SNP | Single nucleotide polymorphism |
| TIA | Transient Ischemic Attack |
| TNF-α | Tumor necrosis factor α |
| T2DM | Type 2 diabetes mellitus |

### Funding

This research was supported by NIH grant R21ES020700 from the National Institute of Environmental Health Sciences, and an unrestricted gift from HOKA ONE ONE.
The funders had no role in study design, data collection and analysis, decision to publish, or preparation of the manuscript.

## Grant Disclosures

The following grant information was disclosed by the authors:
National Institute of Environmental Health Sciences: R21ES020700.
HOKA ONE ONE.

## Competing Interests

The authors declare that they have no competing interests.

## Author Contributions

- Paul T. Williams conceived and designed the experiments, performed the experiments, analyzed the data, prepared figures and/or tables, authored or reviewed drafts of the paper, and approved the final draft.

## Human Ethics

The following information was supplied relating to ethical approvals (i.e., approving body and any reference numbers):

Our analyses of these data were approved by Lawrence Berkeley National Laboratory Human Subjects Committee (HSC) for protocol "Gene-environment interaction vs. quantile-dependent penetrance of established SNPs (107H021)" LBNL holds Office of Human Research Protections Federal wide Assurance number FWA 00006253.

## Data Availability

The data are not being published in accordance with the data use agreement between the NIH National Heart Lung, and Blood Institute and Lawrence Berkeley National Laboratory. However, the data that support the findings of this study are available from NIH National Heart Lung, and Blood Institute Biologic Specimen and Data Repository Information Coordinating Center directly through the website https://biolincc.nhlbi.nih. gov/my/submitted/request/.

Restrictions apply to the availability of these data, which were used under license for this study.

For access to the data, please contact the Blood Institute Biologic Specimen and Data Repository Information Coordinating Center to find information on the human use approval and data use agreement requiring signature by an official with signing authority for their institute. The public summary-level phenotype data may be browsed at the dbGaP study home page.

STATA code is available as a Supplemental File.

## Supplemental Information

Supplemental information for this article can be found online at http://dx.doi.org/10.7717/ peerj.10914#supplemental-information.

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
