# Peer review of "Quantile-dependent expressivity of serum C-reactive protein concentrations in family sets"

_PeerJ, doi:10.7717/peerj.10914_

## Round 0.1 · original submission · Minor Revisions

Please make the suggested corrections, and soon we will have a final decision.

·

Basic reporting

Pass

Experimental design

Pass

Validity of the findings

Pass

Additional comments

I have only one comment: There is no protein in the serum named as "high-sensitivity C-reactive protein". This is wrong. The protein present in the serum is called C-reactive protein (CRP). Researchers often use a highly sensitive technique to measure CRP since the concentration of CRP in the serum is so low that sometimes it cannot be detected without using a sensitive method. However, if a highly sensitive technique is used to measure CRP, it does not change the name of the protein from "CRP" to "high sensitivity CRP". I recommend that, throughout the manuscript, only the term "CRP" is used; all "high sensitivity CRP" should be replaced by "CRP".

Reviewer 2 ·

Basic reporting

- In Line 159 “:” is missing after “Statistics”.
- In Line 209 “.” is missing after the title and there is no separation between the title and the text.
- Table 1: There is no description for the values reported in the parenthesis. In addition, the caption underneath the table is incomplete.
- Figure 2 is best to be labeled as 2.A and 2.B so that the
- In Lines 287, 319, 370, 384, 401, 409, 418, 425, 431, 500, 506, 524, 530, 538, 547 and 552, “.” is missing in the beginning of the lines.

Experimental design

no comment

Validity of the findings

no comment

Additional comments

This manuscript presents a comprehensive study to investigate whether CRP heritability is quantile specific. This is a well written report with thorough analysis and in-depth discussion.

Reviewer 3 ·

Basic reporting

Great job, all suggestions are supporting the study and make it available for publication.
1. I suggest the title of the study be according to "PI/ECOS" criteria (P= population, I= intervention/E= exposure, C= control or compactor, O= outcome, and S= study design), for example, you can check this reference from World Health Organization: https://apps.who.int/iris/bitstream/handle/10665/193040/WHO_HIV_2015.37_eng.pdf?sequence=1
2. The paper requires language editing. For example:
• Line No. 410, and 412: "estradiol" instead of "esterdiol".
• Line No. 520: "12 week" should be "be 12 weeks".
• Line No. 530: you repeated "Ankylosing spondylitis Ankylosing spondylitis" twice.
• Line No. 565: "These exception" should be replaced by "this exception or these exceptions", and other mistakes, so I suggest language editions.
• Besides, you should use unambiguous and comprehensible language.
3. References
It is noted that the argument “Quantile-dependent expressivity” was merely supported by self-citations. Although these citations are relevant and were appropriately used, the author(s) is/are advised to support his/her/their argument with the work of other researchers.

Experimental design

4. I suggest the method section be structured based on Strengthening the Reporting of Observational Studies in Epidemiology (STROBE) Statement: https://www.strobe-statement.org/index.php?id=available-checklists
or as our PeerJ recent publication under the title "The relationship between anti-Müllerianhormone (AMH) levels and pregnancy outcomes in patients undergoing assisted reproductive techniques (AR)", DOI: 10.7717/peerj.10390
5. Table 1. Sample characteristics: Please define in your footnote the following abbreviations: BMI, and CRP

Validity of the findings

6. I suggest the result section be structured based on Strengthening the Reporting of Observational Studies in Epidemiology (STROBE) Statement: https://www.strobe-statement.org/index.php?id=available-checklists

---

## Round 0.2 · accepted · Accept

Great job. Congratulations.

·

Basic reporting

none

Experimental design

none

Validity of the findings

none

Additional comments

none

Reviewer 3 ·

Basic reporting

'no comment'

Experimental design

'no comment'

Validity of the findings

'no comment'

Additional comments

Great job,
Thanks for the language edition, all explanations in method and result sections. So, the article should be accepted.